# Spatiotemporal Patterns of Human–Carnivore Encounters in a Seasonally Changing Landscape: A Case Study of the Fishing Cat in Hakaluki Haor, Bangladesh

**Afsana Nasreen Eva** [1,*] **, Ai Suzuki** [1,2,3] **and Shinya Numata** [1]

1    Graduate School of Urban Environmental Sciences, Tokyo Metropolitan University, Tokyo 192-0397, Japan; ai23suzuki@gmail.com (A.S.); nmt@tmu.ac.jp (S.N.)
2    Graduate School of Asian and African Studies, Kyoto University, Yoshidashimoadachoi-cho 46, Kyoto 606-8501, Japan
3    Research Organization of Open Innovation and Collaboration, Ritsumeikan University, Iwakura-cho 2-150, Ibaraki 567-8570, Japan
*    Correspondence: afsananasreen@gmail.com

**Abstract:** Identifying spatial and temporal patterns of human–carnivore encounters is crucial for predicting conflict hotspots. However, the degree of overlap between human and carnivore movements is likely to differ between stable environments and seasonally changing landscapes. We aimed to clarify key drivers of spatial and temporal overlap of humans and carnivores in a seasonally changing landscape using the case of human–fishing cat encounters in an inland wetland in north-eastern Bangladesh. To obtain encounter information, interview surveys were conducted with 210 respondents in 21 villages in 2020. Monthly rainfall and waterbody size were negatively correlated with the numbers of encounters in the wetland area, while there was no apparent temporal pattern in encounters reported in adjacent villages. Temporal patterns of encounters may be partially explained by human presence (in turn associated with local livelihoods). Except for fishing, intense livelihood activities take place in wetland areas mainly during the dry season. On the other hand, areas peripheral to the wetlands are used for various livelihood activities throughout the year. In a seasonally changing landscape, understanding people's movements could help elucidate spatiotemporal patterns of human–fishing cat encounters at a micro-scale.

**Keywords:** *Prionailurus viverrinus*; seasonally flooded wetland; human movement; livelihood activities; wetland; encounter pattern

## 1. Introduction

Conflicts between humans and carnivores worldwide are becoming a significant concern for conservation scientists and practitioners. Economic losses due to livestock mortality and crop damage, human injuries, and loss of life are the main causes of these conflicts, which often result in the killing of carnivores [1–5]. Hence, human–carnivore conflict is a threat to carnivore conservation. Predicting patterns of conflict [4,6,7] and identifying potential conflict hotspots [5,6] are the initial steps required for effective conservation practice for threatened carnivores.

Spatiotemporal analyses have paid more attention to carnivore ecological aspects, including their population status, hunting strategies [7,8], movement paths, and breadth of habitat use in environmental spaces that they share with humans [4,5,9,10]. However, the perspective of the human dimension is now emerging and as an approach, as it has the importance of effectively understanding the conflict problem that goes beyond the ecological and economic considerations [11]. Researchers have also investigated human aspects such as land use practice within the human-dominated landscape [4,5,10,12]; damage sustained by livestock and crops, loss of human life; the effects of local environmental

settings, including local rainfall patterns [9,13,14] and elevation [9,15]; socio-ecological settings, including attacked livestock farms, villages, and livestock enclosures [5,16,17]; and human population density [5,18,19]. In some studies, conflicts primarily occurred in forest patches and farmlands during the dry season [15]. Other studies found that topographic factors influence spatial patterns of conflict due to variance in human accessibility [10]. Human–carnivore conflicts may also be non-linearly related to human density [20].

Human and carnivore movements are likely to overlap to different degrees between stable environments and seasonally changing landscapes. Human movements and activities are considered to be among the determinants of spatiotemporal patterns in human–carnivore conflicts. Conflict with carnivores is highly likely when people encroach on their habitat in pursuit of their livelihoods. Many human–carnivore conflicts have been examined in relatively stable environments, such as terrestrial and coastal ecosystems. In human–carnivore conflicts, both carnivore and human movements and activities are strongly influenced by the transformation of landscapes that occurs during seasonal inundation [12,21]. Therefore, spatial and temporal patterns of human–carnivore conflicts could vary in a seasonally changing landscape.

In the present study, we focused on spatiotemporal patterns of human–fishing cat encounters (*Prionailurus viverrinus*) in the seasonally changing landscape of the Hakaluki Haor wetland, Bangladesh. Spatial Landsat datasets used in this study shows in Table S1. Hakaluki Haor is a large bowl-shaped tectonic depression that remains underwater during the wet season, but partially dries up in the dry season. Rainfall drives seasonal fluctuation of waterbody size in the haor floodplain. The floodplain is used for agricultural production in the dry season, and particularly supports rice cultivation and fishing activities both in dry and wet seasons [22]. During these livelihood activities, people may encounter many types of wildlife in the wetland and peripheral areas [23]. Therefore, both environmental and anthropogenic factors [20] could directly and indirectly affect the spatiotemporal patterns of human–fishing cat encounters in this seasonally varying wetland ecosystem.

Based on this, we aimed to understand the differences in encounter patterns between the haor and villages within a landscape in which the presence of fishing cats is confirmed in both areas. We hypothesized whether there is a difference in the encounter pattern between the haor and villages within the landscape related to the key drivers of seasonal change, namely rainfall and waterbody size. We focused on monthly rainfall and waterbody fluctuations as environmental factors affecting spatial and temporal patterns of human–fishing cat encounters. As social factors potentially influencing encounter numbers, the population size and elevation of villages were examined. As a potential indicator of conflict, we focused on human and fishing cat encounters, rather than conflict per se, in the wetland and human-dominated areas (villages), because human–carnivore encounters do not always result in conflict and killing of the carnivore, and may frequently go unreported [24]. Understanding encounter patterns could provide insight into how to mitigate conflict with fishing cats. We gathered information on fishing cat encounters in the haor (wetland areas) and villages (human-dominated areas) and examined the general pattern of human–fishing cat encounters, relationships between human–fishing cat encounters in the wetland and seasonal rainfall and waterbody size, and the influence of village population size and elevation on human–fishing cat encounters in villages.

## 2. Materials and Methods

### 2.1. Study Area

The present study was conducted in the large Hakaluki Haor ecosystem, located between latitudes 24°35′ and 24°44′ N and longitudes 92°00′ and 92°08′ E in the north-eastern part of Bangladesh. Due to environmental degradation, Hakaluki Haor was declared an Ecologically Critical Area (ECA) in 1999, and is surrounded by a 40,466 ha environmental protection zone [25]. This haor wetland lies below 9 m a.s.l.; the lower parts of the haor ecosystem, called beels, retain water throughout the year [26]. Hakaluki Haor comprises several hundred beels in the dry season and coalesces to form one huge haor waterbody

during the wet season. The area experiences high annual rainfall (5944 mm in 2017), and is affected by extreme rainfall and runoff caused by flash floods occurring generally during April and May in upstream catchments [22,27].

Approximately 200 villages are clustered along the slightly raised fringes of Hakaluki Haor (Figure 1). People live in the villages adjacent to the haor/beel within the Hakaluki Haor ecosystem. Human settlements extend from the edge of the flooded area up to 40 m a.s.l.. The smallest and largest villages have populations of ~500 and >7000 people, respectively. Villagers use the haor for a variety of activities during the dry season, such as rice production, cattle grazing, fuelwood, reed and grass collection, and harvesting aquatic and other plants.

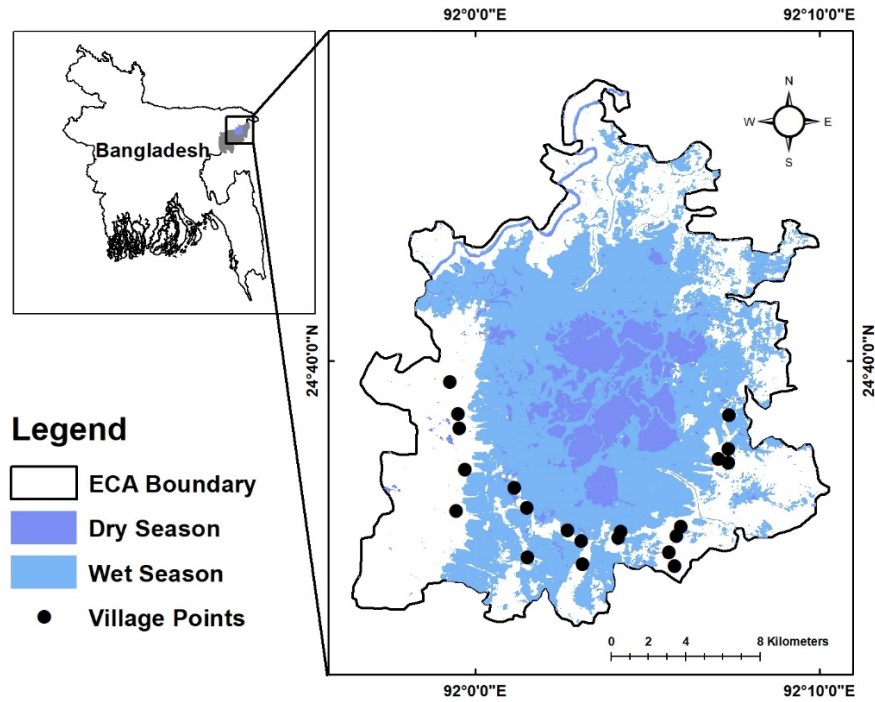

**Figure 1.** The Ecologically Critical Area (ECA) of Hakaluki Haor. Water level differences between the dry season (February 2017) and wet season (September 2017; highest and lowest water levels, respectively).

*2.2. Fishing Cat*

Fishing cats are categorized as Vulnerable (VU) by the IUCN Red List, and as Endangered (EN) on Bangladesh's national red list [28]. Their widely patchy distribution suggests an association with wetlands across the range countries in South and Southeast Asia [29]. Intensified pressure owing to human encroachment speeds the destruction of wetlands greater compared to forests, as wetlands are mostly enclaved inside human habitations [30,31]. Besides habitat loss, killing fishing cats has been also a threat to them due to the depredation of poultry, fish, and other livestock to some degree for consumption and poaching for fur [32,33]. In the case of Bangladesh, people generally perceive fishing cats as tigers and fear encounters with them, and thus, the killing of fishing cats by local people has become one of the main concerns for their long-term survival [34].

In Hakaluki Haor, fishing cats are observed within the fragmented patch of wetland vegetation, beside beels, surrounding ponds, and sometimes inside mosques and abandoned houses in villages. Villagers show strong motivation to kill fishing cats [35]. Considering the overlap between people and fishing cats, it is an urgent issue to understand when and where people encounter fishing cats. Although we need to investigate fishing cat movement within the seasonally changing landscape to grasp the whole picture of human–fishing cat encounters, at the same time, we can start taking actions towards

reducing encounters on the human side without waiting for ecological information on fishing cats in the landscape.

### 2.3. Collection of Human–Fishing Cat Encounter Data

A semi-structured questionnaire was designed to investigate fishing cat encounters, both spatially and temporally. A pilot survey was conducted in north-eastern Hakaluki Haor (30 participants) to test the questionnaire. After finalizing the questionnaire based on the pilot survey, we surveyed 21 villages using the purposive sampling method; 210 participants were drawn from the area surrounding Hakaluki Haor, which has a population density of 500–900 people per km$^2$ [36]. The survey was conducted in February and March 2020, and 4 years of encounter data were obtained for the period 2016–2019. We did not consider a longer time frame because the details and timing of earlier encounters would likely be recalled with less precision.

The male and female respondents were aged ≥15 years and included farmers, fishermen, local transport drivers, businesspeople, bricklayers, carpenters, schoolteachers, religious teachers, housewives, and unemployed persons.

All interviewees were residents of Hakaluki Haor who had lived in the area for at least 10 years. The interviews took approximately 15–30 min and were conducted in a comfortable environment. We introduced ourselves and our objectives, and then described the interview process (e.g., the types of questions, and the participants' right not to answer or stop the interview at any time), stressing that there would be no legal consequences for sharing information [33,37]. Before proceeding to the interview, the interviewees were asked to describe the physical characteristics of fishing cats (e.g., head, body, legs, tail), and the differences between fishing cats and three sympatric carnivores: the jungle cat (*Felis chaus*), golden jackal (*Canis aureus*), and small Indian civet (*Viverricula indica*).

The questionnaire asked whether the interviewees had previously encountered fishing cats. Those who had had an encounter then provided their age and occupation, and described when (year, month, and season) and where (forest, poultry house, pond, or elsewhere) the encounter took place, the activity that they were engaged in at the time, how they acted during the encounter, and the outcome (i.e., whether the cat was killed, injured, rescued, or escaped). They also provided information about anyone else involved in the encounter and indicated who could validate the encounter. The locations of encounters in villages were determined by GPS and then photographed. The locations of encounters in haor areas could not be precisely determined because the respondents refused to travel to the haor due to the large distance involved.

### 2.4. Environmental and Social Dataset

To analyze the spatial and temporal patterns of human–fishing cat encounters, we particularly focused on two water-related variables driving seasonal change in the landscape, namely rainfall and the size of waterbodies. Data on rainfall for 2016–2019 were collected from the Bangladesh Meteorological Department. Monthly waterbody size was determined from 2016–2019 remote sensing data obtained from Landsat 8 (OLI) and Landsat 7 (ETM+) satellite images (Table S1). Unfortunately, it was impossible to collect data for June 2016 and July 2017 due to excessive (100%) cloud coverage throughout the study area. Therefore, we used the average waterbody size for those months. We processed the images and calculated the water level in ArcGIS 10.4.1 (ESRI, Dhaka, Bangladesh) using the Modified Normalized Difference Water Index (MNDWI) [38]. The MNDWI is expressed as follows:

$$MNDWI = \frac{(Green - SWIR)}{(Green + SWIR)} \tag{1}$$

where Green represents pixel values from the green band and SWIR denotes pixel values from the short-wave infrared band.

To understand patterns of village encounters according to social characteristics, we obtained human population data for each surveyed village from the Bangladesh Bureau of

Statistics Population Census [36]. Village elevation was determined based on GPS during interview surveys in residential areas. Seasonal livelihood activities were derived based on interviewees' reported activities in haor and village areas to understand how encounters with fishing cats related to seasonal activities.

*2.5. Data Analysis*

We assessed the comparison of season and times of encounter between the haor and village using Fisher's exact test. We also analyzed the temporal relationships of monthly rainfall and waterbody size (for the period 2016–2019) with the number of human–fishing cat encounters, in both the haor and village areas, using Spearman's rank correlation tests. Temporal relationships between the month of encounter and distance from the closest haor waterbody were analyzed using the Kruskal–Wallis test. All analyses were performed using R statistical software (version 4.2.0; R Core Team, Vienna, Austria).

**3. Results**

*3.1. General Patterns of Encounters with Fishing Cats*

In total, 63 interviewees reported encounters with fishing cats, while 147 reported no encounters. Respondents of all ages experienced encounters (15–20 years, 6%; 21–30 years, 13%; 31–40 years, 24%; 41–50 years, 25%; 51–60 years, 22%; >60 years, 10%). The majority of the interviewees who had encountered fishing cats were farmers (52%), followed by fishermen (19%), businesspeople (16%), those with other occupations (8%), housewives (3%), and the unemployed (2%). A total of 75 encounters were reported by interviewees between 2016 and 2019 (Figure 2). Among these 75 encounters, 11 resulted in the killing of the fishing cat; thus, 64 encounters were non-lethal. Encounters occurred in both the haor (64%) and village (36%) areas. There were 6 encounters in 2016, 14 in 2017, 18 in 2018, and 37 in 2019.

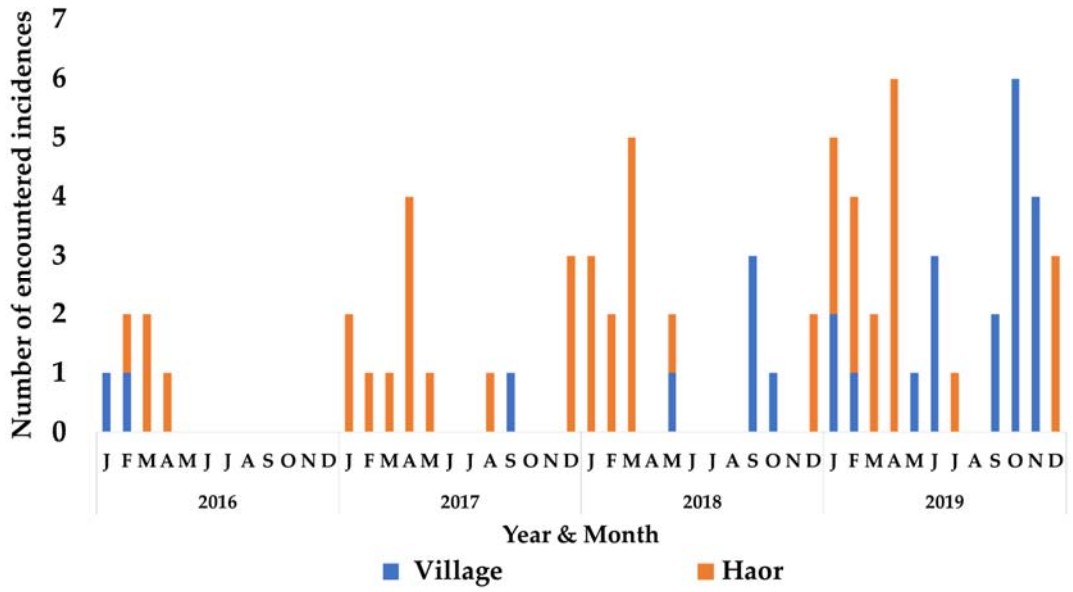

**Figure 2.** Monthly variation in human–fishing cat encounters in village (n = 27) and haor (n = 48) areas during the period 2016–2019.

*3.2. Temporal Patterns of Encounters*

Temporal patterns of encounters with fishing cats differed between the haor and village areas, despite their close proximity. We found significant differences in times of encounter (dawn, daylight, dusk, and night) between the haor and village (Fisher's two-tailed test, $p = 0.002$). There were 34 encounters at dusk, 18 during the day, 17 at dawn, and 6 at night (Table 1). The number of encounters was the highest at dusk in the haor (n = 27),

while no clear peak in encounters at any particular time was observed in villages (Table 1). Most of the haor encounters occurred during the dry season (December to April). During the wet season, there were also encounters in villages (Figure 2).

**Table 1.** Difference in time and activity engaged during encounters between the haor and village areas. Both daytime and nighttime encounters were reported in villages. Diurnal encounters dominated in the haor.

| Occupations of Villagers Encountering Fishing Cat (s) | Haor | | | | | Village | | | | |
|---|---|---|---|---|---|---|---|---|---|---|
| | | Times of Encounter | | | * Activity Engaged in during the Encounter | | Times of Encounter | | | * Activity Engaged in during the Encounter |
| | Dawn | Daylight | Dusk | Night | | Dawn | Daylight | Dusk | Night | |
| Farmer | 1 | 10 | 22 | - | a | 4 | 1 | 5 | 2 | c |
| Businessman | 3 | - | - | - | b | 1 | 4 | 2 | - | d |
| Fisherman | 7 | - | 5 | - | b | - | - | - | - | - |
| Gardener | - | - | - | - | - | - | 1 | - | 1 | e |
| Carpenter | - | - | - | - | - | - | 1 | - | 1 | f |
| Teacher | - | - | - | - | - | - | - | - | 1 | g |
| Housewife | - | - | - | - | - | 1 | - | - | 1 | h |
| Unemployed | - | - | - | - | - | - | 1 | - | - | i |
| **Total** | **11** | **10** | **27** | **-** | | **6** | **8** | **7** | **6** | |

* a—farming, cattle grazing, duck rearing, fuelwood collecting; b—fishing; c—crossing road, crossing pondside road; d—crossing road, crossing pondside road, entering mosque; e—tea gardening; f—working in an abandoned house, crossing road; g—entering mosque; h—household chores; i—traveling.

The temporal patterns of encounters associated with rainfall and waterbody size differed between the haor and villages. We found significant relationships between the number of monthly encounters and both monthly rainfall and estimated water body size (Figures 3 and 4). Monthly encounter numbers were negatively correlated with total monthly rainfall ($r_s = -0.32$, $p < 0.05$) and total waterbody size ($r_s = -0.52$, $p < 0.01$) within the haor (Figure 4a,c). However, no significant correlation was found between the village encounter numbers and rainfall ($r_s = -0.11$) or waterbody size ($r_s = -0.09$).

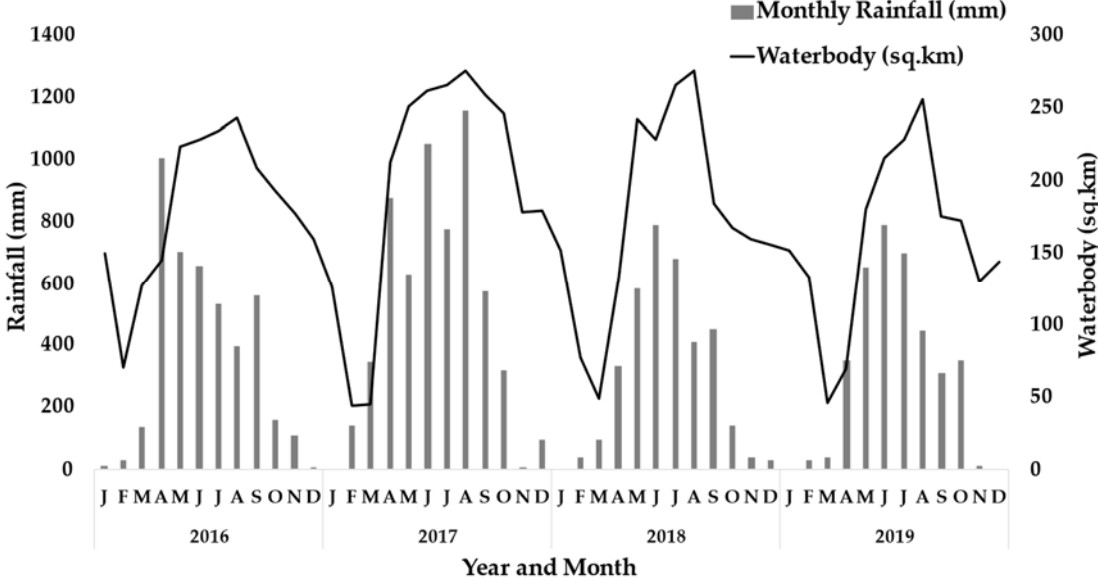

**Figure 3.** Changes in monthly rainfall and waterbody size within Hakaluki Haor.

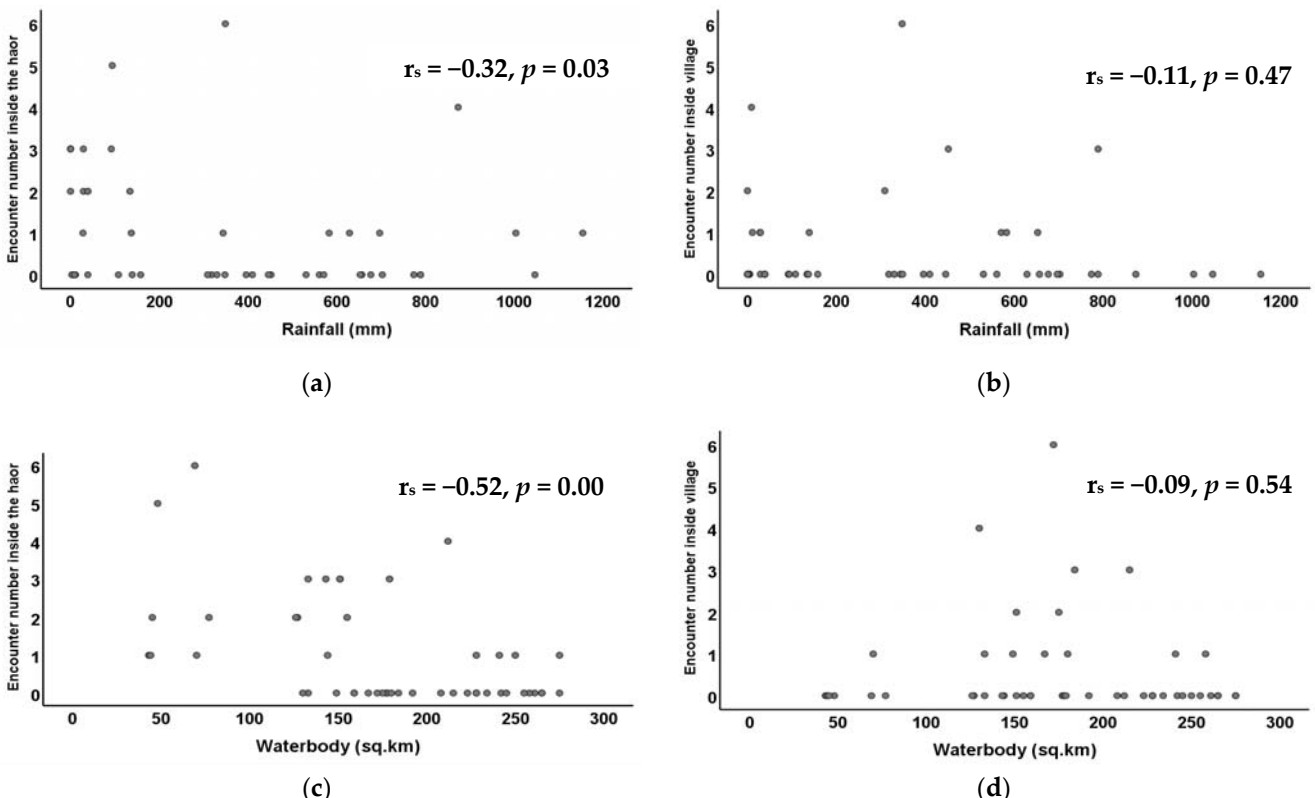

**Figure 4.** Correlations between encounter numbers and monthly rainfall and waterbody size. (**a**) Encounter numbers inside the haor were negatively correlated with rainfall while (**b**) no significant correlation was found between village encounter numbers and rainfall. Likewise, (**c**) encounter numbers inside the haor were negatively correlated with waterbody size but (**d**) no significant correlation was found between village encounter numbers and waterbody size.

*3.3. Spatial Patterns of Encounters*

No clear spatial pattern of encounters was observed in the village data; encounter numbers was not significantly related to elevation ($r_s = 0.31$, ns) or village population size ($r_s = -0.11$, ns) (Figure 5). Furthermore, the distance between the encounter and the closest haor waterbody did not vary significantly throughout the year (Kruskal–Wallis test, $p = 0.11$).

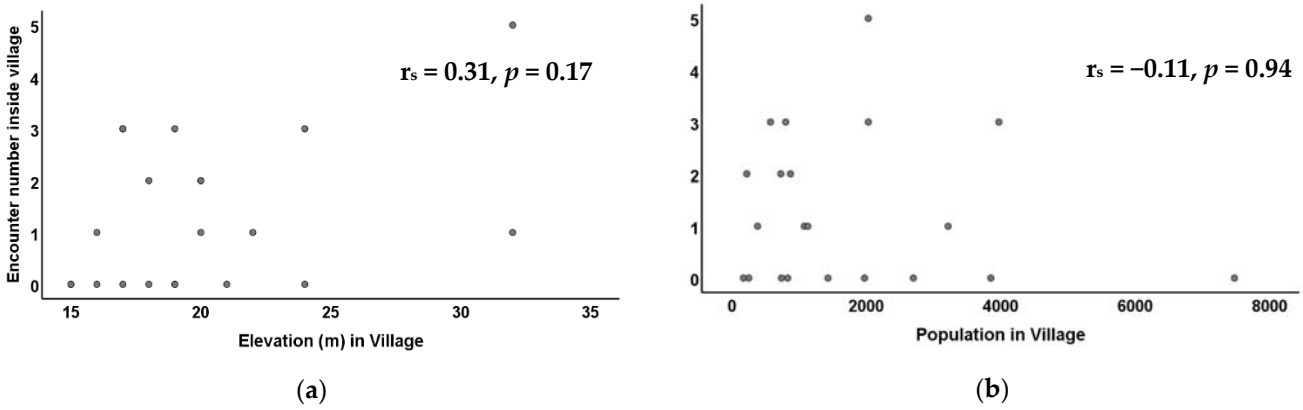

**Figure 5.** Relationships between encounter numbers and village (**a**) elevation and (**b**) population size.

*3.4. Seasonality in Encounters during Livelihood in the Haor and Village Areas*

There were more encounters during the dry ($n = 53$) than wet season ($n = 22$). There were highly significant seasonal encounter differences (dry and wet seasons) between

the haor and village (Fisher's two-tailed test, *p* < 0.0001). Within the haor, we observed seasonality in local livelihoods. During the dry season, villagers used various locations within the haor for different livelihood activities (Table 2). Some villagers traveled daily to forest patches within the haor to collect fuelwood, while others traveled to grasslands to graze cattle, and transformed the edges of haor waterbodies into paddy fields. Fishing was the only activity that took place both in the dry and wet seasons. Nearly half of all reported encounters occurred during crop production activities, and 35% of encounters took place while villagers were fishing. Many encounters occurred close to forest patches (44%, n = 21), in agricultural fields (29%, n = 14), and in areas adjacent to waterbodies (27%, n = 13). During the wet season, fewer agricultural activities took place in the haor because land used for agriculture or grazing was flooded, and fishing was the main activity.

**Table 2.** Locations and months of human–fishing cat encounters in Hakaluki Haor.

| Area | Locations of Encounter | Season of the Use for Livelihood | Dry | | | | Wet | | | | | | Dry | |
|---|---|---|---|---|---|---|---|---|---|---|---|---|---|---|
| | | | 1 | 2 | 3 | 4 | 5 | 6 | 7 | 8 | 9 | 10 | 11 | 12 |
| Haor | Cropping Field | Dry | X | X | | X | X | | | | | | | X |
| | Forest | Dry | X | | X | | | | | | | | | X |
| | Grassland | Dry | | | X | | | | | | | | | X |
| | Waterbody | Both | | | X | X | | | X | X | | | | |
| Village | Road | Both | X | | | | X | | | | X | X | X | |
| | Pond | Both | | X | | | X | X | | | | | X | |
| | Abandoned House | Occasionally | X | | | | | | | | | | | |
| | Mosque | Both | | X | | | | X | | | | | | |

X denotes the presence of encounters with fishing cats within an area.

People resided in villages throughout the year. Village-based encounters were mostly related to farming (44%, n = 12), other occupations (30%, n = 8), and business activities (26%, n = 7). Village encounters occurred close to bushy patches and vegetation adjacent to roadways (33%), in graveyards (19%), near poultry and livestock enclosures (11%), ponds (15%), mosques (7%), abandoned houses (4%), and in tea gardens and paddy fields (11%).

## 4. Discussion

In this study, encounters between people and fishing cats were not uniformly distributed across the area of interest. In Hakaluki Haor, encounter patterns differed between the haor and village areas. While encounter numbers peaked during the dry season in the haor, no such pattern was observed in villages. The timing of encounters also differed, particularly at night. Within the haor, no nighttime encounters were reported, whereas encounters occurred in villages both during the day and at night.

Notably, the outcomes of encounters also differed between the haor and village areas. In the haor, encounters with fishing cats tended to be distant sightings that seldom resulted in killing. Within villages, encounters with fishing cats can occur at any time of year, either as distant sightings or close encounters. The sudden shock of close encounters led some people to attempt to catch and kill fishing cats; these attempts occasionally succeeded.

Seasonal differences in human–fishing cat encounters could be potentially explained by the variance in human presence with seasonal rainfall patterns and changes in waterbody size. Encounter numbers in the haor were negatively correlated with rainfall intensity; encounters were more common in less rainy months. The haor encounters were also negatively associated with water level. In the haor during the wet season, we found that waterbody size increased fivefold from its dry season size. However, during the dry season, the haor areas dry up to reveal extensive areas of green land, except in low-lying depressions that still contain water [26]. During the wet season, more than 200 km² of dry land is inundated. This significant seasonal change in available land triggers more intensive use of the haor in the dry season. Previous studies also reported the livelihood activities that take place in the haor during the dry season, such as rice cultivation, poultry farming, cattle grazing, day laboring (in brick and crop fields), and fishing in waterbodies persisting in low depressions [22,39]. These seasonal livelihood activities reflect a larger

human presence in the dry season. Consequently, the probability of human–fishing cat encounters increases during the dry season relative to the wet season.

In contrast to the seasonal patterns of encounters observed in the haor, no significant correlations were found in villages according to rainfall intensity or waterbody size. Although areas surrounding villages are submerged during the wet season (Figure 1), the numbers of encounters in villages did not reflect seasonal variations in the local environment. People encountering fishing cats in villages had various occupations, and most of them encountered fishing cats while walking or traveling within the village, rather than during livelihood activities. Such daily travel takes place throughout the year and shows little seasonality.

The probability of conflict varies according to people's seasonal movements and use of the land. A study conducted on the Pantanal wetland of Brazil found that encounters increased during the dry season due to overlap between species and human use of seasonally flooded grasslands [23,40]. A previous study dealing with large carnivore species found that a number of attacks primarily took place during the dry season in forests and farmlands [15]. In addition to seasonality, our findings also suggest that daily patterns of encounters could be predicted based on people's movements. People generally enter and use the haor wetlands for livelihood purposes between dawn and dusk [41], except for some duck rearing and fishing activities in the dry season that involve building small tents or housing to stay over the season. Consequently, no nighttime encounters were recorded in the haor in this study, while encounters in the night were reported in villages.

Caution should be exercised when attributing the differing encounter patterns between the haor and village areas. Fishing cats' movements and use of the landscape also display seasonality [42]. In Hakaluki Haor, fishing cats may move from flooded areas towards higher ground near the water's edge, where villages are located. In that case, seasonal movements of people and fishing cats are likely to be similar, leading to large spatial overlap. However, fishing cats' presence in higher elevations throughout the year is not clear.

The villagers in this study encountered fishing cats near dense vegetation patches, bushy vegetation with ponds, mosques, and abandoned houses (Table 1 and Figure 6). Fishing cats may actively use such places in human-dominated areas. Seasonal movements of fishing cats in the Hakaluki Haor require further study.

Based on the findings of the present study, we conclude that human–fishing cat encounters in the seasonally changing landscape of Hakaluki Haor are dynamically related to villagers' use of the area, and their responses to seasonal landscape changes. Even at distances within 5 km, daily and seasonal encounter patterns and the likelihood of conflict can differ. We suggest that this is probably not only due to species movement, but also people's livelihood activities, and the use of the area could serve as predictors in the seasonally changing landscape, particularly when seeking to understand the details of such encounters.

Therefore, our findings suggest that micro-scale livelihood activities could constitute an important factor in the heterogeneity of encounter patterns within a seasonally changing landscape. When the killing of the species is observed at a concerning rate and there is a need to take immediate action, this approach could help to develop strategies by identifying where and when encounters can happen based on how people move throughout and use the seasonally changing landscape. Needless to say, the ecological survey of the species is fundamental and crucial to fully understanding the encounter patterns to develop effective conservation action.

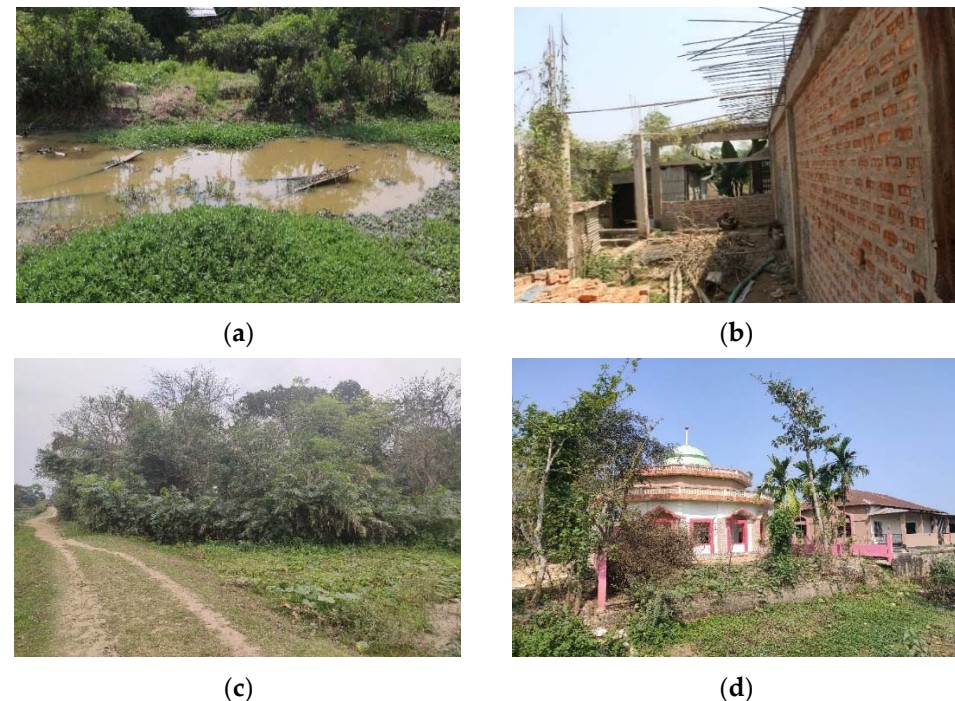

**Figure 6.** Examples of village environments where human–fishing cat encounters have occurred: (**a**) pond surrounded by vegetation, (**b**) abandoned house, (**c**) road adjacent to bushy patch, (**d**) mosque surrounded by vegetation. Photographs by Eva A. N.

**Supplementary Materials:** The following supporting information can be downloaded at: https://www.mdpi.com/article/10.3390/conservation2030027/s1, Table S1: Spatial Landsat datasets used in this study.

**Author Contributions:** Conceptualization, A.N.E., A.S. and S.N.; Data curation, A.N.E., A.S. and S.N.; Formal analysis, A.N.E., A.S. and S.N.; Investigation, A.N.E.; Methodology, A.N.E., A.S. and S.N.; Supervision, A.S. and S.N.; Validation, A.N.E., A.S. and S.N.; Writing—original draft, A.N.E.; Writing—review and editing, A.N.E., A.S. and S.N. All authors have read and agreed to the published version of the manuscript.

**Funding:** The fieldwork was funded by the "JSPS KAKENHI" program, grant number 19K20490 and supported by the "Tokyo Human Resources Fund for City Diplomacy (THRFCD)" as a scholarship from Tokyo Metropolitan University. The funder had no role in the study design, data collection and analysis, decision to publish, or preparation of the manuscript.

**Institutional Review Board Statement:** Ethical review and approval were waived for this study due to Tokyo Metropolitan University (TMU) does not have clear criteria for ethical approval in non-sensitive issues.

**Informed Consent Statement:** This study does not deal with data about illegal activities, sensitive information that may arise ethical issues, experiments using animals, medical health data, or the use of genetics or biological information.

**Data Availability Statement:** Not applicable.

**Acknowledgments:** The authors would like to acknowledge the scholarship received from Tokyo Metropolitan University and JSPS KAKENHI grant program. We appreciate the Bangladesh Meteorological Department for sharing the meteorological data. Moreover, we would like to thank Mohammad Abdul Aziz, (for the support and advice in the field), Mohammad Nayeem Aziz Ansari, (for developing the initial concept) and Ummeh Saika, (for giving technical support) from Jahangirnagar University, Bangladesh. We are thankful to the field members for their volunteer support in data collection and technical support in mapping analysis. We appreciate the core participants of the surveyed community for giving their valuable time in our data collection interview procedure. Last

but not least, we are grateful to all our laboratory colleagues for providing information and input necessary comments to make the study a success.

**Conflicts of Interest:** The authors declare no competing interests.

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
