# Peer review of "Spatiotemporal Patterns of Human–Carnivore Encounters in a Seasonally Changing Landscape: A Case Study of the Fishing Cat in Hakaluki Haor, Bangladesh"

_conservation, doi:10.3390/conservation2030027_

Round 1
Reviewer 1 Report
Manuscript ID: conservation-1769758
Title: Spatiotemporal patterns of human-carnivore encounters in a seasonally changing landscape: A case study of the fishing cat in Hakaluki Haor, Bangladesh
Authors: Afsana Nasreen Eva *, Suzuki Ai, Numata Shinya
Review
In general, nice and simple manuscript, definitely worth to publish, but having some problem to be solved before acceptance.
First main comment: information on the fishing cat is lacking. Why this species is important? What is it’s status? Starting point might be IUCN page, https://www.iucnredlist.org/species/18150/50662615 and a few more references. At an international journal like Conservation, definitely not all readers
Second, critical comment is: insufficient statistical treatment of data. Correlation is not enough. In the text you are using percentages – that is, proportions. Proportion, as a rule, should have confidence interval (95% CI would be ok). For comparison of proportions there is good method, G-test. It is available in R (package RVAideMemoire), or as online calculator.
Third comment – please make clear, it is Haor or haor, and how you compare this to villages (aren’t villages part of Hakaluki Haor?
Fourth comment: is it “human and carnivore conflict”in case of fishing cat? I understand encounters may be a threat to species, but not conflict to humans. Please add information and references, if I am not correct.
Then there are several more comments:
Abstract
Lines 13–18: too long as introduction in limited size of Abstract. When shortening, please find a place for 1 sentence on the species status (VU, decreasing, limited and fragmented distribution?).
Keywords: add Prionailurus viverrinus; delete “human and carnivore conflict”
Introduction
Line 79: haor, (wetland area) – so it seems, haor is habitat, and Haor is place name. Please make sure reader will understand difference and how you use these two words in the text. So far it is not clear.
Line 90: masl – maybe better “m a.s.l.”?
Material and methods
Figure 1. The ecologically Critical Area – capitalize second word
Line 117: interviewees – maybe, respondents?
2.2 should be merged with 2.3 under the subtitle “Collection of human-fishing cat encounter data”
2.5 – as said above, add statistical methods
Results
Lines 170–180: add CI for proportions and use G-statistics to tell, if there are significant differences. Percent is not enough.
Table 1: please test, if proportions, based on the numbers in the Table, are significant, and show significant differences with superscript letter added to the values. I also propose to code activities by letters in the footer, and present only letters to make Table more tight.
Figure 2: use real colors, journal do not charge for color figures. Also please test, if there are differences in number of encounters between months; if yes – note these months in the Figure and explain in the caption. If no difference, add statement to the caption.
Figure 4: regression line should be added to all four graphs. Caption, Lines 205–207 – there are mistypes of wrong punctuation used. Check Y axis title, why capital letters are used for “Number”; Haor or haor – you mean place or habitat?; (c) and (d) should be in bold.
Figure 5: regression line should be added; Check Y axis title, why capital letters are used?
Table 2: could you use “Fit to contents” function to make column 2 fit in one line?
Lines 223–226. 229–234: check G-statistics, if differences exist, and add CI to values
Figure 6: (c) and (d) should be in bold.
Parts of the Figures 4, 5 and 6 are not of the same size and are not aligned.
Back matter
Supplementary Materials: consider Table S1: Spatial Landsat datasets used in this study instead of Table S1: The following is a summary of the spatial Landsat datasets that were used in this study.
Informed Consent Statement: add, it is necessary
Conflicts of Interest: Any role of the funders in the design of the study; in the collection, analyses or interpretation of data; in the writing of the manuscript, or in the decision to publish the results must be declared in this section. If there is no role, please state “The funders had no role in the design of the study; in the collection, analyses, or interpretation of data; in the writing of the manuscript, or in the decision to publish the results”.
References
Line 385: add doi: 10.3390/ani2040591
[31] and [32] are different publications? Just checking
Author Response
''Please see the attachment''

Reviewer 2 Report
Thank you for allowing me to review this manuscript. When I first read the title of this work, I was excited about the study. The topic is extremely interesting, as it provides an example of a medium-sized carnivore species and the conflicts it may induce. This is a special case, little is known here. I was, however, quite disappointed when reading the text in detail, as I get the feeling that the biological foundation is missing.
Most importantly, the authors should develop a clear hypothesis and goal. Looking at encounter rates without reflecting on potential kinds of conflict does not really add a lot of value.
The authors promise to present „spatio-temporal patterns” of fishing cats and they insinuate that these data help to "predict conflict hotspots" but, in my opinion, the study lacks a clear hypothesis and clear objectives. The study surely has potential but the authors should conduct major revisions before this manuscript reaches the status of deserving publication in this journal.
Line 38-40: „Spatiotemporal analyses have been paid more attention to carnivore ecological aspects including their population status, hunting strategies, movement paths, and breadth of habitat use in environmental spaces that they share with humans.“
When you say “more” it should be clear what does this “more” refer to exactly… More than what? Please clarify.
Line 71-77: Authors state that they use human-fishing cat encounters as an indicator of conflict but it remains uncertain, what kind of conflict they refer to. The quality of the introduction could be increased, if the authors briefly outline some thoughts about the concept of “conflict” and the type of conflict that might be induced by the fishing cat in their study. Authors might pay attention and reflect on the fact that perceiving a conflict through the lens of humans – in an anthropocentric context – does not always do justice to the complexity of the underlying problem. Sometimes, more ecocentric perspectives are.
Figure 4 needs adjustments: the labelling of a, b, c, and d does not match with the sections of the figure. The x-axes have no label – please clarify!
Line 276-277: “People are primarily attacked during the dry season in forests and farmlands.” This sentence needs justification and interpretation by the authors. I mean: A small cat of about 8kg that is mainly feeding on rodents, other small mammals and fish attacks people… I`d like to know more about these attacks…
Additional questions / aspects emerge that should be addressed in the manuscript:
- Please develop clear hypothesis to be tested and clear objectives to be addressed – what is the goal of your study and what can we learn for other areas or in terms of human-carnivore conflict in general?
- Please include statements about the feeding habits and the ecology of the fishing cat and please describe what kind of conflict these medium-sized mammals might induce for humans and then compare and interpret the perceptions of the interviewed people
- In how far could the results of the encounter rates be influenced by the nocturnal lifestyle of the fishing cat?
- Can you develop recommendations for action? Can you predict or suggest areas of potential high conflict?
Author Response
''Please see the attachment''

Round 2
Reviewer 1 Report
I accept revision with only few comments:
Point 4: Fourth comment: is it “human and carnivore conflict” in case of fishing cat? I understand encounters may be a threat to species, but not conflict to humans. Please add information and references, if I am not correct.
Response 4: Carnivore can threaten people’s safety and livelihoods, which can lead to conflict resulting in people killing animals in self-defence or retaliatory killing. But conflict may not be necessarily posed always threats to humans. In the case of Bangladesh, people are perceived fishing cat as tiger and feel fear to encounter.
perhaps you could add the last sentence " In the case of Bangladesh, people are perceived fishing cat as tiger and feel fear to encounter " to the manuscript
Table 1, footer, use:
* a – farming, cattle grazing, duck rearing, fuelwood collecting; b – fishing; c – crossing road, crossing pondside road; d – crossing road, crossing pondside road, entering mosque; e – tea gardening; f– working in an abandoned house, crossing road; g – entering 218 mosque; h – household chores; i – travelling.
Figure 4: please clean remains of the formet symbpols in upper right corners
Caption of figure 4: please check with Editor for the correct placement of (a), (b), (c) and (d); also, clear mistypes
Line 259: test should be named: (Fisher's two-tailed test, p < 0.0001)
Author Response
''Please see the attachment''

Reviewer 2 Report
Dear authors,
thank you for a careful revision of the manuscript. I think that the study increased in quality. However, there are (at least) two issues that should be addressed before the study should be considered for publication: Please again check Figure 4: There are strange numbers/characters occuring on the upper left side of the sub-figures a ("3"), b ("`"), and d ("4") - could you please remove this?
Secondly, based on my previous comment, you hav now added the following sentence: "Previous study showed number of attacks primarily
took place during the dry season in forests and farmlands [17]."
Besides the fact that there are too many spaces after the word "place", this sentence needs clarification. Your discussion is about the fishing cat and then you immediately talk about large carnivore attacks without mentioning that you are talking about large carnivores - please clarify. You could say something like: Studies dealing with large carnivore species found that ....
Thank you
Author Response
''Please see the attachment''
